# Characteristics of the Thermal Environment and its Guidance to Ecological Restoration in a Resource-Based Area in the Loess Area

**DOI:** 10.3390/ijerph20043650

**Published:** 2023-02-18

**Authors:** Shihan Liu, Dandan Wang, Yingui Cao

**Affiliations:** 1School of Land Science and Technology, China University of Geosciences, Beijing 100083, China; 2Key Laboratory of Consolidation and Rehabilitation, Ministry of Natural Resources, Beijing 100035, China

**Keywords:** thermal environment, mining and reclamation, profile method, cooling effect, compound area of mine agriculture urban

## Abstract

The thermal environment is a crucial part of ecological environments. It is vital to study the distribution and generation of thermal environments for regional sustainable development. Mining area, agricultural area and urban area were taken as the research object, and remote sensing data were used to study the spatiotemporal distribution characteristics of the thermal environment. The relationship between the thermal environment and land use types was analyzed, and the effect of mining and reclamation on the thermal environment was emphasized. The main findings were: (1) the thermal effect zone in the study area was dispersed. The area ratio of the thermal effect zone accounted for 69.70%, 68.52%, 65.85%, 74.20% and 74.66% in the year 2000, 2003, 2009, 2013 and 2018, respectively. The contribution to the overall thermal effect was in the order of agricultural area > mining area > urban area. (2) The proportion of forest and the average grid temperature always showed a significant negative correlation in different scales and had the highest correlation and the greatest influence. (3) The land surface temperature (LST) of opencast areas was higher than the surrounding temperature, and the temperature difference was 3–5 °C. The LST of reclaimed sites was lower than the surrounding temperature, and the temperature difference was −7 to 0 °C. The quantitative study found that reclamation mode, shape and spatial location could affect the cooling effect of the reclaimed site. This study can provide a reference for the mitigation of thermal effects and the identification of influences of mining and reclamation on the thermal environment in the coordinated development of similar regions.

## 1. Introduction

As coal is China’s main energy source, the coal industry is a pillar of China’s economy. With the advancement of industrialization and urbanization, some cities rely on the mining and processing of coal resources to promote regional economic development. Over time, the development of coal resources in resource-based areas has brought significant consequences to the local environment [1], including various environmental problems such as microclimate changes and environmental pollution. As an important ecological and environmental problem, the thermal effect has received more and more attention in resource-based areas [2,3,4]. However, it will form a compound area that includes mining area, agricultural area and urban area with the expansion of mines [5]. The coordinated development of mining area, agricultural area and urban area is strongly linked to human survival and development, which is the key to regional sustainable development. The thermal environment is an extremely important part of the ecological environment, which is related to regional sustainable development.

Since the heat island effect was formally proposed, researchers have conducted in-depth studies on the spatial distribution, formation mechanisms and mitigation measures of the thermal environment using conventional meteorological observations, remote sensing (RS) technology and numerical simulations. RS technology is widely used in research of the thermal environment because of good dynamics, wide coverage and low acquisition cost [6,7,8]. Appropriate data and methods are critical to ensure the effective use of RS image information [9]. According to the characteristics of different thermal infrared data, researchers have come up with a variety of land surface temperature retrieval algorithms, including single-channel algorithms and multi-channel algorithms [10,11,12]. As one of the important environmental factors, the land surface temperature (LST) retrieval algorithms have been deeply studied and applied in resource-based areas. There are distinctions in the suitability of the algorithms for different study areas and application fields. Many researchers have compared the accuracy of different LST retrieval algorithms focusing on resource-based areas and found that the mono-window algorithm has higher accuracy [8,13]. They have also attempted to improve the recognition accuracy of the underlying surface to obtain a more accurate LST retrieval result [14,15]. In fact, the LST in resource-based areas tends to be higher than in other areas due to special land use types such as opencast areas.

Revealing the spatiotemporal distribution characteristics of the thermal environment is the basis of thermal effect research. Many scholars have described thermal characteristics by using qualitative descriptions, thermal landscapes, the gravity center model and more [16,17,18,19]. Identifying the factors influencing the thermal environment is a prerequisite for proposing mitigation measures. The influencing factors have been analyzed in depth from the macro to the micro scale, from singleness to comprehensiveness, from feature determination to quantity determination and from universality to typicality. Land use types are described as the most significant human disturbance to the environment, and it is an important factor influencing the thermal environment in resource-based areas. It is mainly studied from two aspects, including scattered land use types [2,20,21] and continuous ground parameters [3,22,23]. At the scale of mining areas, industrial and mining land under development often has the highest LST, while stripped areas, reclaimed sites and water areas have lower LSTs [2,3]. At the scale of the town or city, bare land, built-up land, industrial land and mining land have higher LSTs [22,23], while water areas have the lowest LST [23]. The compound area of mine agricultural urban is an interdependent community of resources, economy and society [5]. Mining areas, agricultural areas and urban areas often have different types and amounts of land use. It is important to compare and analyze the influence of land use types on the thermal environment in different functional areas from multiple perspectives, as this has great influence on sustainable development of resource-based areas.

For the same land use type, the LST still shows a large variation, and the influence on the thermal environment is also different; this is due to the heterogeneity of internal characteristics (e.g., shape characteristics) [4]. There are more and more studies focusing on the spatiotemporal variations of the thermal environment and landscape pattern as well as the influence of the composition and configuration of landscape patterns at different scales [24,25,26]. These studies have several conclusions. First, industrial and mining land is the main factor causing an increase in LST, and its area, shape and configuration affect the thermal environment [4,27,28,29]. The larger the patch area of industrial and mining land, the higher the LST [4,27,29]. The more complex the patch shape of industrial and mining land is, the lower the LST and the slower the heating rate [28]. The more patches are clustered in industrial and mining land, the higher the LST [28]. Second, green land and water areas have a good slow-release ability for thermal effects, which is usually quantitatively represented by average LST, cooling range and other indicators. The cooling effect of green land and water areas is related to landscape characteristics and spatial layout, such as area, shape, aggregation index and greenness [30,31,32,33]. These studies focused on the influence of industrial and mining land, green land and water on the thermal environment. However, studies on the influence of mining and reclamation on the thermal environment are rare. In particular, the impact of the landscape pattern of reclaimed sites on the thermal environment remains unclear. It is of great significance for ecological restoration to clarify this effect.

The Pingshuo coal mine has advanced the coal industry practice, encouraging regional economic development. However, high-intensity mining activities can cause damage to the originally fragile ecosystem [34]. As an important indicator of the ecological environment, it is urgent to study the thermal environment in resource-based areas. Therefore, the thermal environment of mining area, agricultural area and urban area in the Pingshuo coal mine area in the past 20 years was studied with the following objectives: (1) reveal the contribution of different functional areas to thermal effects and the influence of land use types on the thermal environment in the different functional areas of resource-based areas; (2) quantify the range of influence of mining and reclamation on the thermal environment and the mechanisms of influence of the landscape pattern of reclaimed sites on the thermal environment. Finally, the realization of these objectives will contribute to the proposal of management countermeasures for mining development, agricultural production, urban construction and reclamation.

## 2. Materials and Methods

### 2.1. Study Area

The study area is situated in Pinglu District, Shuozhou City, Shanxi Province of China (111°52′ E–112°41′ E, 39°21′ N–39°58′ N). It covers an area of 517 km^2^ on the Loess Plateau and has a complex and diverse topography (Figure 1). Loess is widely distributed in the area, with sandy soil texture and low water content. The zonal vegetation in Pinglu District is dry grassland with sparse vegetation. The study area belongs to the temperate continental monsoon climate zone with four distinct seasons, concentrated rain and heat in the summer. The average annual precipitation is about 450 mm, and the annual evaporation is between 1786.1 mm and 2598 mm. Drought and wind damage frequently occur in the spring, and the spring drought in 2013 severely affected agricultural production.

The research area has three open-pit mines called Antaibao, Anjialing and East. It is the most economically developed and mining-intensive area in Pinglu District. It has great influence on the regional urban development, coal industry progress and agricultural production.

### 2.2. Data Sources

#### 2.2.1. Remote Sensing Data

The Landsat data from USGS (https://earthexplorer.usgs.gov/, accessed on 16 March 2022) were used to retrieve LST and vegetation coverage data. The data (Table 2) were basically collected between June and July when the vegetation was luxuriant. Cloud-free images were selected to truly reflect the ground energy distribution and change trend. The accuracy of the temperature retrieval results was cross validated by comparing them to the Landsat series LST data from PSC APP (https://15203878955lz.users.earthengine.app/view/psc-app, accessed on 22 June 2022). In situ experiments showed that the total root-mean-square error (RMSE) at seven sites is 2.01 K [35].

#### 2.2.2. Meteorological Data

The near-surface air temperature data at satellite overpass times obtained from the “China meteorological forcing dataset (1979–2018)” [36,37] were used as effective mean atmospheric temperature inputs for Landsat LST retrieval. The data were in NETCDF format, with 3 h in temporal resolution and 0.1° in horizontal spatial resolution. The data accuracy was between the meteorological administration observation data and the remote sensing data (http://data.tpdc.ac.cn, accessed on 16 March 2022).

### 2.3. Methods

#### 2.3.1. Data Processing

Radiometric calibration, atmospheric correction and image mosaic processing were performed using ENVI5.3 software. Other operations, including land use classification and vegetation coverage extraction, were also implemented in ENVI5.3 software. Based on the Landsat data and Google Earth high-resolution images, the land use types were divided into eleven categories: farmland, forest, grassland, urban land, rural settlement, transportation land, industrial site, opencast area, stripped area, dumpsite and water. We used ArcGIS10.6 software to establish a 2 × 2 km grid based on the land use classification results in 2018 so as to study the quantitative relationship between LST and land use types. We also used a 500 × 500 m grid for analysis and found that the positive and negative characteristics of the results did not change; only the absolute values were different, which would not have a significant impact on the results. Therefore, a total of 101 grids completely located in the study area were extracted based on the 2 km scale, and the average LST of each grid and the area proportion of each land use type were determined.

The vegetation coverage in the study area was estimated based on *NDVI* using the sub-pixel model [38]:(1)$$P_{V}=\frac{NDVI-NDVI_{S}}{NDVI_{V}-NDVI_{S}}$$
where *NDVI_S_* and *NDVI_V_* are the *NDVI* of bare soil and vegetation, respectively, which have values of 0.05 and 0.70 [39]. *NDVI* is the value of each pixel.

#### 2.3.2. LST Retrieval and Thermal Environment Classification

The mono-window algorithm was used to retrieve the LST from Landsat data [10]. The estimation process follows Equation (2).
(2)$$\begin{array}{c} T_{S}=\frac{\left\{a\left(1\;-\;C\;-\;D\right)\;+\;\left[b\left(1\;-\;C\;-\;D\right)\;+\;C\;+\;D\right]T_{sensor}\;-\;DT_{a}\right\}}{C}-273.15 \end{array}$$
where *a* and *b* are the regression coefficients, whose value depends on the band and the temperature range [10,12]. *T_sensor_* is the at-sensor brightness temperature; *T_a_* is the effective mean atmospheric temperature calculated by the acquired near-surface air temperature [40] and *C* and *D* are the intermediate parameters given by:(3)C=ετ
(4)$$D=\left(1-\tau \right)\left[1+\left(1-\varepsilon \right)\tau \right]$$
where *ε* is the ground emissivity. Since the excavated bottom of the opencast is mainly covered with coal, different emissivity values of 0.925, 0.940 and 0.985 were set according to previous studies [41,42,43], and different temperature retrieval results were compared with the Landsat series LST to determine the actual emissivity to calculate the ground emissivity. *τ* is the atmospheric transmittance calculated by the atmospheric water vapor content. The MOD05 data obtained from NASA (https://ladsweb.modaps.eosdis.nasa.gov/search/, accessed on 16 March 2022) was used to retrieve the atmospheric water vapor content, which was substituted into the equation [12,39] to calculate the atmospheric transmittance.

To make the spatiotemporal distribution characteristics of the thermal environment comparable at different satellite overpass times under different weather conditions, the LST retrieval results were first normalized from zero to one and then divided into seven categories using the mean–standard deviation method. The temperature range for each category is shown in Table 3. In this study, medium temperature, sub-high temperature, high temperature and extremely high temperature were included in the thermal effect zone, and the other three grades were included in the non-thermal effect zone [44].

#### 2.3.3. Spatiotemporal Distribution Characteristics of the Thermal Environment

In order to further reveal variations in the spatial distribution of the thermal effect zone in the past 20 years, the gravity center model was used to measure the spatial transfer of the gravity center of thermal effect zones in different functional areas [18,19].

#### 2.3.4. The Influence of Typical Areas on the Thermal Environment

Typical areas were defined as the area which may influence the thermal environment due to mining and reclamation, and this included three opencast areas (ATB, AJL and DLT) and nine reclaimed sites (ATB_N, ATB_X, ATB_XK, ATB_NS, ATB_NX, NSG, AJL_N, AJL_D and AJL_X), which were vectorized in ArcGIS10.6 software according to Google Earth high-resolution images. The distributions of typical areas are shown in Figure 1, and the meaning of abbreviations is displayed in Table 1.

The influence range, amplitude and rate of typical areas on the thermal environment in mining area were studied based on the profile method [29]. To avoid the randomness of a single profile line, two or more profile temperature rays were selected around each typical area. The influence amplitude (∆T) was calculated as the difference between the temperature of the typical area and the surrounding land:(5)$$\Delta T=T_{b}-T_{a}$$
where ∆T is the influence amplitude; T_a_ is the temperature of the typical area (which is the LST in the initial state of the profile line) and T_b_ is the surrounding temperature (which is the value where the LST changes drastically or reaches a relatively stable position).

The influence range (L) is the distance to T_b_. The influence amplitude and influence range of the typical area jointly determine its influence rate (I), which is calculated by Equation (6):(6)$$I=\frac{\Delta T}{L}$$

## 3. Results

### 3.1. Validation of the LST Retrieval Results

To match the scale of Landsat series LST, the LST retrieved from Landsat in this paper was resampled to 100 m by calculating the mean LST in each 100 m grid cell. Then, the RMSE of the Landsat series LST and Landsat LST in the whole study area were compared. The results revealed that the RMSE of the two data sources is below 1.4 °C in all cases, as shown in Figure 2. The absolute difference can be caused by the differences in temperature retrieval algorithms and resampling methods [45]. At the same time, it can be seen that the ground emissivity affects the accuracy of the LST retrieval. Regardless of the year, the RMSE of the two data sources was minimum when emissivity was 0.985. Therefore, the emissivity was finally determined to be 0.985. In addition, the mean value of the RMSE in each year was less than 1 °C, so the retrieval results reflected the distribution of the LST in the study area and can be used for further analysis.

### 3.2. Spatiotemporal Distribution Characteristics of the Thermal Environment

Figure 3 illustrates the spatiotemporal distribution characteristics of the thermal environment, Figure 4 demonstrates the land use classification result and Figure 5 explains the vegetation coverage distribution. Generally, the non-thermal effect zone (as defined in Section 2.3.2) was basically kept in the areas with high vegetation coverage, water coverage or reclaimed coverage. In contrast, the thermal effect zone was dispersed in the study area, and the change characteristic was irregular. The area ratio of the thermal effect zone accounted for 69.70%, 68.52%, 65.85%, 74.20% and 74.66% in the year 2000, 2003, 2009, 2013 and 2018, respectively, and it displayed a similar fluctuating rising trend as the average land surface temperature [46]. Different functional areas have different land use types, which causes different effects on the thermal environment. Figure 6 evaluates such impacts in 2018 from the perspectives of the proportion of thermal effect zones and the average LST, with similar results for other years. The thermal effect zone of the mining area basically distributed in opencast areas, industrial sites, dumpsites and cropland. In the agricultural area, the thermal effect zone mainly distributed in cropland and grassland. Although the average LSTs of industrial sites and opencast areas were higher, they occupied a small area and can be almost ignored. The thermal effect zone of the urban area mainly distributed in cropland and industrial sites.

According to the gravity center model, the displacement change and aggregation state of the thermal effect zone were obtained (Figure 7). The gravity center of the thermal effect zone in the agricultural area had a trend of north first and then south, with large changes in the early stage and relative stability in the later stage. The transfer law of the gravity center of the thermal effect zone in the mining area was similar to that in the agricultural area, and the transfer trend to the east was more obvious due to the impact of mining. The gravity center of the thermal effect zone in the urban area showed the trend of south first and then north. The gravity center of the thermal effect zone in the whole study area was basically consistent with that in the agricultural area. Based on the above analysis, it can be seen that the contribution to the overall thermal effect was in the order of agricultural area > mining area > urban area.

### 3.3. Influences of Land Use Types on the Thermal Environment

The relationship between LST and land use types was quantitatively analyzed by establishing fishnets. Table 4 shows the data that passed the significance test. In the whole study area, the proportion of grassland showed a significant positive correlation with the average grid temperature. In the mining area, the proportion of opencast area and dumpsites were significantly positively correlated with the average grid temperature, and the former had a stronger warming effect. The proportion of water area was negatively correlated with the average grid temperature, but the influence was much greater than other land use types; however, the fitting effect is difficult to comprehensively summarize the regional situation due to the small number of grids of water area. In the agricultural area, the proportion of grassland is positively correlated with the average grid temperature. In general, the proportion of forest and the average grid temperature showed a significant negative correlation at different scales and had the highest correlation and the greatest influence.

### 3.4. Influences of Typical Areas on the Thermal Environment

Mining and reclamation can affect the regional microclimate and then influence residents’ lives and change phenology. It is of great significance to effectively identify their influences on the thermal environment.

#### 3.4.1. Influence of Mining on the Thermal Environment

Based on the LST retrieval results in 2018, three opencast areas were selected as typical areas to analyze the influence of mining on the thermal environment. Figure 8 was formed by averaging the temperature profile lines around the typical areas. It can be seen that the temperature of opencast areas is higher than the surrounding temperature (Figure 8a). Additionally, the influence amplitude was between 3 °C and 5 °C. Moreover, the influence range was between 0.1 and 0.5 km, and the order of influence rate was AJL > ATB > DLT. By comparison, it was found that the average temperature of each typical area was in the order of AJL > ATB > DLT. The AJL and ATB with concentrated disturbances had higher average temperatures than that of DLT.

#### 3.4.2. Influence of Land Reclamation on the Thermal Environment

Based on the LST retrieval results in 2018, the influence of nine different reclaimed sites on the thermal environment were compared to analyze their reclamation effects. It can be seen that the temperature of the reclaimed site was lower than the surrounding temperature (Figure 8b). Additionally, the influence amplitude was between −7 °C and 0 °C. Moreover, the influence range was within 0.6 km, and the order of influence rate was ATB_X > NSG > ATB_NX > ATB_N > AJL_N > AJL_D > ATB_NS > ATB_XK > AJL_X.

This study selected six indexes (Table 5) representing the size, shape, reclamation mode, reclamation term, spatial location and composition of surrounding land use types and then analyzed the influence of the characteristics and external conditions of the reclaimed site on its cooling effect.

To exclude the influence of other land use types with high rates of human disturbance on the cooling effect, the profile lines through cropland, forest or grassland were selected to analyze the relationship between the above six indexes and the average temperature (T_m_), influence range (L), influence amplitude (∆T) and influence rate (I) of reclaimed sites.

The correlation analysis was conducted for six indexes to eliminate redundant factors before analyzing the relationship between each index and the cooling effect of the reclaimed site, and the results are shown in Figure 9. The statistical analysis found that D had a significant positive correlation with Y and S, and S had a significant inverse association with PD. As the Antaibao opencast and the Anjialing opencast gradually pushed east, the dumpsite far away from the advancing direction of mine was the first to complete the dumping work, and the reclamation had basically finished. However, the extent of reclamation of the nearby dumpsite was smaller under the influence of mining and dumping, and the reclamation work started later. The reclaimed sites with smaller areas were simultaneously close to other dumpsites and industrial sites. Therefore, the correlations between indexes were spurious. As a result, the above six indexes were retained, and the equation with the best fitting effect was selected to quantitatively describe their influence on the cooling effect of the reclaimed site in SPSS software.

T_m_ had a significant negative correlation with *P_V_* (Table 6). This shows that the reclamation mode influences the LST. In other words, reclamation into the forest and grassland has a higher vegetation coverage than the reclamation into cropland (such as NSG and ATB_NS), and the complete reclamation into the forest and grassland has a higher vegetation coverage than the partial reclamation into them (such as ATB_X and ATB_N). As a result, the former has a relatively low average temperature, and the regression equation is shown in Table 7. There was a significant positive association between ∆T and *P_V_*, and the regression equation is shown in Table 7. The higher the vegetation coverage is in the reclaimed site, the better cooling effect it has, meaning it will have a lower average temperature (i.e., the initial state temperature of the profile line) and bigger influence amplitude. ∆T was significantly positively correlated with Y, but the relationship between them cannot be summarized simply. There was a significant inverse association between ∆T and FRAC, and the regression equation is shown in Table 7. The more complex the boundary shape of reclamation land is, the smaller the influence amplitude is. ∆T and D showed a significant positive correlation, and the regression equation is shown in Table 7. The cooling influence amplitude of the reclaimed site decreased first and then increased with the distance from the gravity center of the thermal effect zone.

## 4. Discussion

### 4.1. Spatiotemporal Distribution Characteristics and Factors Influencing the Thermal Environment

Due to the particularity of land use types in the study area, this study discussed the emissivity value of opencast areas. By comparing with the Landsat series LST, an emissivity relatively consistent with the actual value was found to calculate the LST, which could reflect the distribution of LSTs in the study area. Nevertheless, it is necessary to study a fast, accurate and in-depth LST retrieval method suitable for mining areas.

Relevant studies have shown that opencast areas, industrial sites and dumpsites have high LSTs due to exposed coal and gangue, frequent human activities and increased soil compaction [2]. However, few studies have found higher temperatures in cropland and grassland. This research found that most of the cropland and grassland is located on the slopes of hillsides and presents a high temperature state. Due to the low vegetation coverage and large soil exposure, the land surface receives more solar radiation. Moreover, the soil in the study area is loose and dry, so the surface heats up quickly [47]. In the past 20 years, the average LST shows a fluctuating rising trend of decreasing first and then increasing, which is consistent with the background of a significant increase in summer LST on the Loess Plateau [46].

Fishnets were established to quantitatively analyze the influence of land use types on the LST, and it was found that the warming effect of the opencast area is more significant [2]. The vegetation coverage of forests is relatively high, which can provide shade for the ground, absorb heat through evapotranspiration and reduce the LST [48]. Therefore, the proportion of forest land and the average grid temperature showed a significant negative correlation at different scales and had the highest correlation and the greatest influence. However, this paper only used linear regression equations to analyze the influence of different land use types on the thermal environment. Beyond that, the thermal effect zone was scattered and had irregular changes, which may also be affected by topographical factors. At the same time, only the influence of land use area on the thermal environment was considered in this study. Hence, the analysis of influencing factors such as topography and spatial location of land use types can be supplemented in future work. In the future, other mathematical equations or models can be used to analyze the specific degree of influence of different factors under the premise of adding multiple influencing factors.

### 4.2. Influences of Mining and Reclamation on the Thermal Environment

Mining development has aggravated the high-temperature aggregation effect of the thermal environment [10,23], and the LSTs of opencast areas are 3–5 °C higher than the surrounding temperature. Post-mining reclamation activities can effectively improve the regional climate [49,50], and the LSTs of reclaimed sites are 0–7 °C lower than the surrounding temperature. Since the mining status is related to the actual resource conditions and mining technique, and because the boundary of the opencast area is blurred, the influence of mining on the thermal environment needs to be explored accurately and quantitatively.

Compared with previous research on industrial and mining land [4,29], there are new findings. Similar to our results, these studies found that industrial and mining land had an effect on the thermal environment, and its temperature was higher than that of the surrounding land. Identifying and quantifying the cooling effect of reclaimed sites will benefit land reclamation in future. In addition to analyzing the influence of opencast areas on the thermal environment, this paper also analyzed the influence range of reclaimed sites on the thermal environment and the influence mechanism of landscape patterns on the thermal environment. Relevant studies have shown the influence of dumping technology and reclamation attributes on the LST of reclaimed sites [50,51]. In this study, six indexes representing the characteristics and external conditions of reclaimed sites were selected to analyze their influence on the thermal environment. There was a significant inverse association between Tm and *P_V_*. The LST decreased by 1.63 °C for each 0.1 increase in vegetation coverage. Except reclamation to cropland, the vegetation coverage of the reclamation to forest and grassland was basically between 0.5 and 1, which can effectively play a cooling role. There was a significant positive correlation between ∆T and *P_V_*. The increase in vegetation coverage lowered the LST effectively, while the difference in the surrounding temperature was small, which increases the influence amplitude. As the vegetation coverage increases, the influence amplitude changes more. ∆T and Y had a significant positive correlation, but the relationship between them cannot be simply summarized due to various interferences on the reclaimed site [52]. The more complex the boundary shape of the reclaimed site is, the closer the interior and exterior heat exchange is, and the stronger the influence on thermal environment is. As a result, ∆T had a significant negative correlation with FRAC, and Tm had a positive correlation with FRAC. At the same time, the interior of the reclaimed site is more disturbed by the thermal environment, so its cooling effect is slowed down to a certain extent [30]. ∆T and D had a significant positive correlation and presented a quadratic relationship. That is, the influence amplitude of reclaimed sites decreased first and then increased with the distance between the gravity center of the temperature of each typical area and the gravity center of the thermal effect zone of the mining area. The closer the distance to the gravity center of the thermal effect zone, the reclaimed site and the surrounding land influenced by mining to a greater extent, and all had higher LSTs. The reclaimed site was still influenced and had a high temperature when the distance increased. However, the profile lines were oriented away from the gravity center of the thermal effect zone, and the surrounding land was basically unaffected with a lower temperature. When the distance increases steadily, the reclaimed site and the surrounding land will be minimally affected, and the temperature will be low and in a normal state.

Due to the particularity of the mining and reclamation layout, there are correlations among the indexes, and the legacy effects of mining should be quantified [53]. Consequently, the multiple regression equations cannot be established, and the degree of influence of different indexes is difficult to reflect. The multi-directional profile lines obtained to increase the number of samples may solve this problem. In addition, due to the influence of data accuracy and acquisition, further research should be carried out in the future, such as research aimed at increasing the resolution of the supervised classification results to analyze the cooling effect of the interior characteristics of reclaimed sites and comprehensively considering the influence of the vegetation configuration in the horizontal and the vegetation structure in the vertical on the thermal effect.

### 4.3. Countermeasures and Suggestions for Thermal Environment Management

For the study area, it is important to improve the soil exposure status, continuously carry out land reclamation and stabilize the urban expansion rate. Regarding the size of the thermal effect and the contribution to the overall thermal effect, it is in the order of agricultural area > mining area > urban area. In the future, it will be necessary to adhere to the principle of adapting measures to local conditions and to continuously carry out ecological restoration projects. In order to ensure the quantity of cropland and grassland in the region, attention should be paid to the improvement of cropland quality to avoid the random growth of grassland caused by cropland abandonment, and attention should be paid to the improvement of grassland vegetation coverage to avoid large areas of soil exposure. Many studies have shown that developing vegetation can effectively improve surface biophysical characteristics [54,55], and the cooling effect of vegetation is also affected by the vegetation type [56]. The contribution of mining area to the thermal effect was lower than that of agricultural area, which may benefit from land reclamation. In the future, the concepts of reclamation along with mining and ecological production should continue to be kept [57,58]. For the industrial site with a fixed position, measures such as increasing the green rate can be taken to alleviate the thermal effect. In order to restrain the negative impact of mining on the thermal environment, it is of great importance to regularly monitor and maintain the reclamation effect. Such monitoring facilitates the implementation of prevention, protection and restoration methods and can even be rolled out globally [55]. The urban area had the lowest contribution to the thermal effect. The urban land is basically in the non-thermal effect zone, yet there has been a transitional trend to the thermal effect zone in recent years. The urban expansion mode and land use pattern should be reasonably controlled in the future. In the studies of the urban heat island effect, different measures such as green roofs, cool pavements and geometric structures of urban buildings have also received attention [59].

For the mining area, it is important to reasonably plan the layout of the reclaimed site and consider the reclamation effect in the long term. The disturbance range of the mining area will converge into larger ones, and the disturbance intensity will change from weak to strong. It is difficult for a single reclaimed site to alleviate this phenomenon, and a reclaimed site may even lose its cooling effect due to strong interference. The cooling effect of reclaimed sites is affected by various factors. The more complex the shape of the reclaimed site, the more likely it is to be disturbed by the outside environment [30], and the influence amplitude of the cooling effect will be smaller. Considering the actual conditions of mining, the reclaimed site is suitable to be planned as a regular shape. The reclamation mode of forest or grassland can effectively reduce the average LST of reclaimed sites and increase the influence amplitude of the cooling effect. The farther the reclaimed site is from the gravity center of the thermal effect zone in the mining area, the bigger the influence amplitude of the cooling effect. Therefore, dumpsites near opencast areas can be considered to be reclaimed into forests or grasslands to try to control the thermal effect generated by mining to a small spatial extent. In contrast, the reclamation to cropland should be placed in a remote position for an optimal reclamation effect. As one of the contents of ecological effect evaluation, the cooling effect of the reclaimed site includes various identification factors such as the influence range, amplitude and rate, and there are many influencing factors. In future land reclamation work, the most representative identification factor can be selected according to the actual situation for regular monitoring and maintenance.

## 5. Conclusions

This study took the compound area of mine agricultural urban areas in the Loess region as the research object. The spatiotemporal distribution characteristics of the thermal environment were illustrated, the relationship between land use types and the thermal environment in different functional areas were revealed and the influence of mining and reclamation on the thermal environment were discussed. The following conclusions can be reached from the results:(1)The thermal effect zone is scattered and has irregular changes, while the non-thermal effect zone is basically maintained in areas with high vegetation coverage, water coverage or reclaimed coverage. The proportion of the thermal effect zone shows the same fluctuating rising trend as the average land surface temperature. According to the gravity center model of the thermal effect, the contributions of different functional areas to the overall thermal effect of the study area are in the order of agricultural area > mining area > urban area. However, urban land has tended to transform into the thermal effect zone in recent years, so the urban expansion mode and land use configuration should be reasonably controlled in the future.(2)The LST can be affected by land use types. The average grid temperature has significant correlations with the proportion of forest, grassland, opencast areas and dumpsites in different research scales. However, the proportion of forest and grid average temperature show a significant negative correlation with the highest correlation and the greatest influence. Thus, it is important to ameliorate the soil exposure status and improve vegetation coverage according to local conditions.(3)Mining and reclamation influence the warming effect and cooling effect, respectively. In general, continuous work and regular monitoring of land reclamation are of great significance. Specifically, in future reclamation processes, it can be considered to plan the reclaimed site as a regular shape, select the reclamation mode of forest and grassland and try to reclaim as forests and grasslands near opencast areas and reclaim as croplands far away from the opencast area. Due to the influence of resource endowment and mining conditions, the boundary of opencast areas is blurred. Therefore, it is necessary to further quantify the influence of mining on the thermal environment in future.

## Figures and Tables

**Figure 1 ijerph-20-03650-f001:**
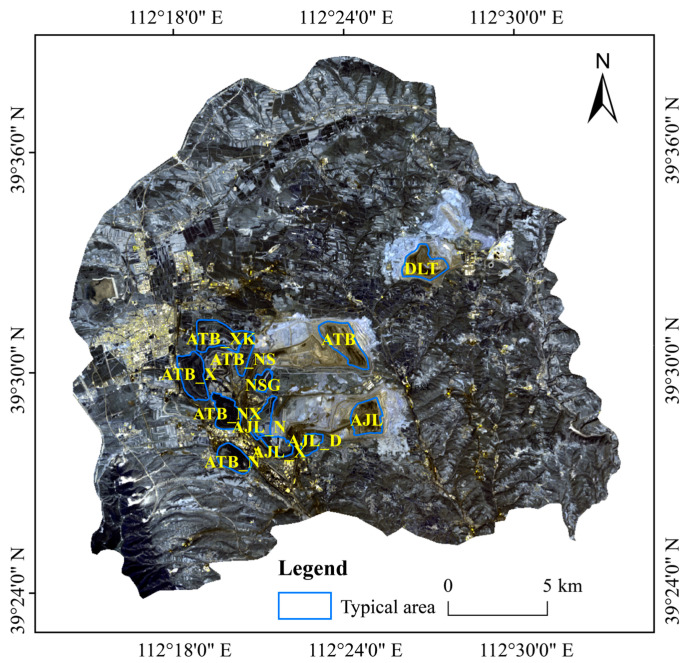
Study area and typical areas from true-color Landsat 8 image in 2018. See Table 1 for the full name of typical area abbreviations.

**Figure 2 ijerph-20-03650-f002:**
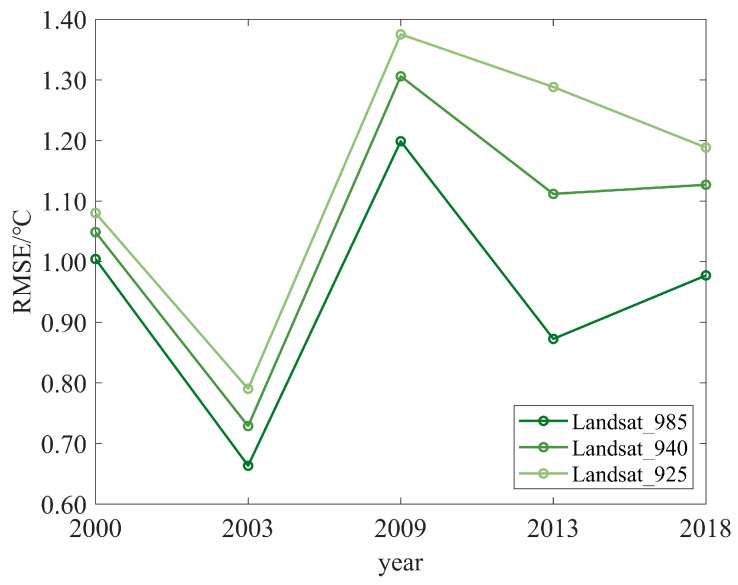
The RMSE of Landsat series LST and Landsat LST. Landsat_985, Landsat_940 and Landsat_925 represent the value of LST when the ground emissivity of opencast areas is 0.985, 0.940 and 0.925, respectively.

**Figure 3 ijerph-20-03650-f003:**
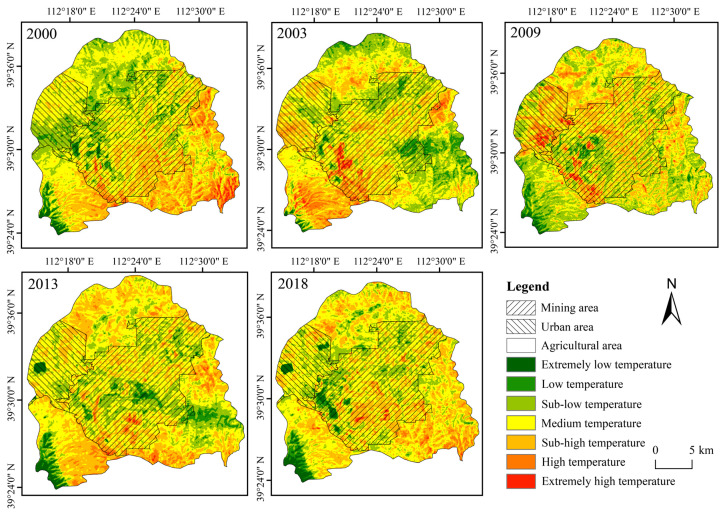
Thermal environment distribution and change in different functional areas.

**Figure 4 ijerph-20-03650-f004:**
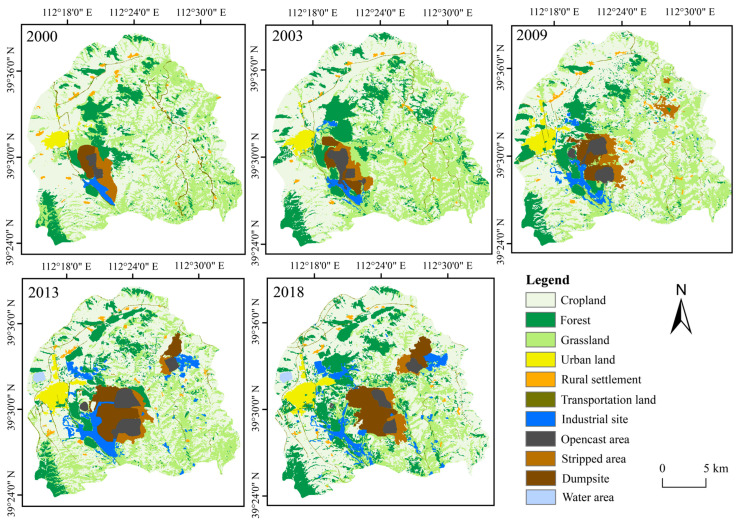
Land use distribution and change in different functional areas.

**Figure 5 ijerph-20-03650-f005:**
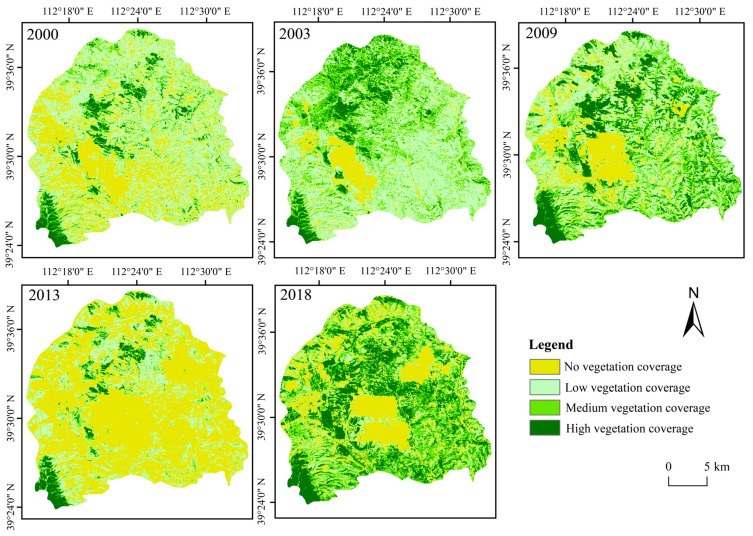
Vegetation coverage distribution and change in different functional areas.

**Figure 6 ijerph-20-03650-f006:**
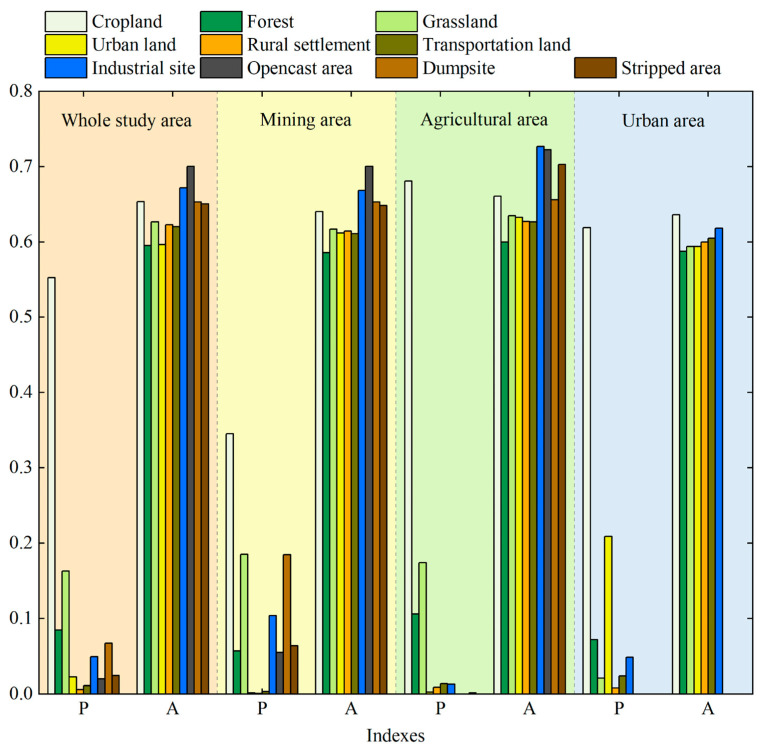
Impacts of different land use types in different functional areas on the thermal environment. P represents the proportion of the thermal effect zone, and A shows the average LST.

**Figure 7 ijerph-20-03650-f007:**
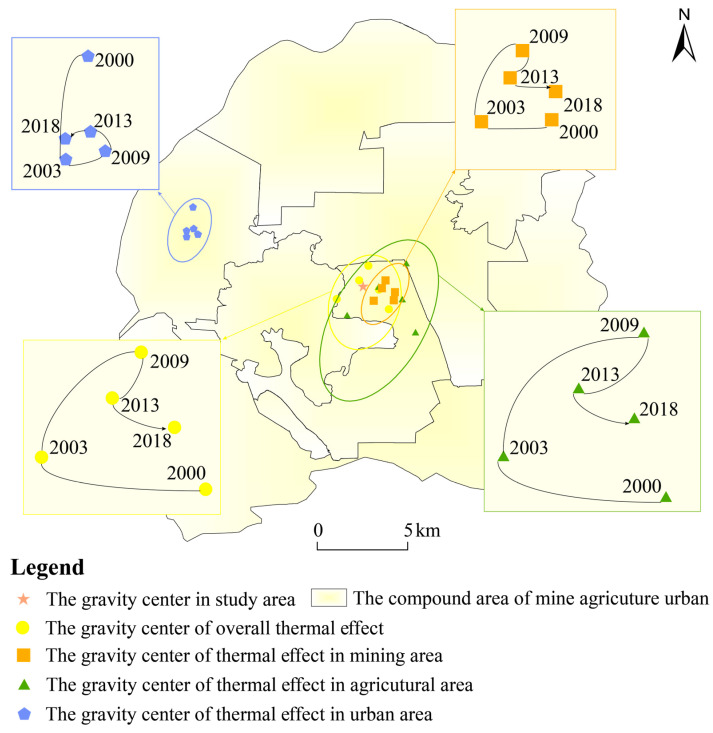
The gravity center of the thermal effect zone and its changes in the study area.

**Figure 8 ijerph-20-03650-f008:**
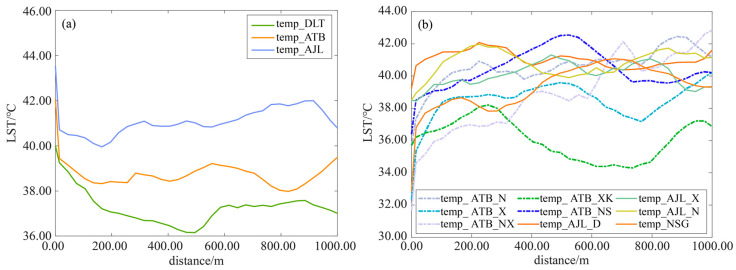
Variation characteristics of LST around different typical areas in mining area (**a**) and reclaimed sites (**b**). See Table 1 for the full name of typical area abbreviations.

**Figure 9 ijerph-20-03650-f009:**
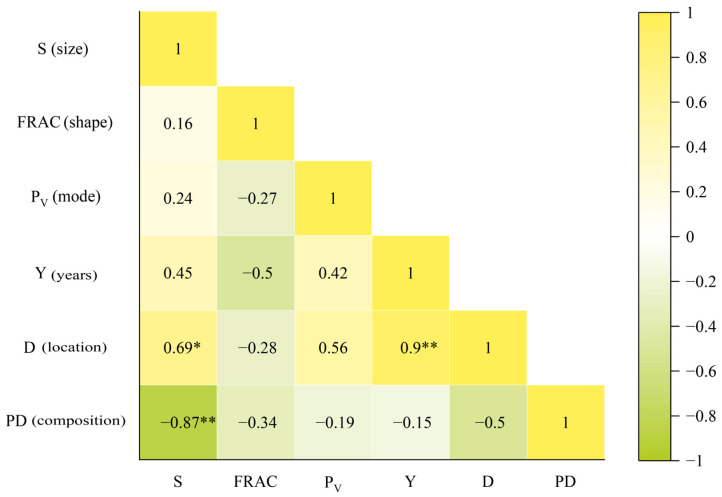
Results of correlation analysis among six indexes * *p* < 0.05; ** *p* < 0.01.

**Table 1 ijerph-20-03650-t001:** The meaning of abbreviations.

Abbreviations	Meaning
DLT	Opencast in East open-pit mine
ATB	Opencast in Antaibao open-pit mine
AJL	Opencast in Anjialing open-pit mine
ATB_X	West dumpsite in Antaibao open-pit mine
ATB_N	South dumpsite in Antaibao open-pit mine
ATB_XK	West expansion dumpsite in Antaibao open-pit mine
ATB_NS	Upper internal dumpsite in Antaibao open-pit mine
ATB_NX	Lower internal dumpsite in Antaibao open-pit mine
NSG	Nansigou dumpsite in Antaibao open-pit mine
AJL_X	West dumpsite in Anjialing open-pit mine
AJL_N	Internal dumpsite in Anjialing open-pit mine
AJL_D	East dumpsite in Anjialing open-pit mine

**Table 2 ijerph-20-03650-t002:** Details of the remote sensing data.

Data Source	Date	Time (Hour: Minute)	Extent	Spatial Resolution (m)	Temporal Resolution (day)
Landsat 5	30 June 2000	10:48 10:49	Path 126/Row32 and Path 126/Row 33	30/120	16
	9 July 2003	10:48 10:48			
	9 July 2009	11:00 11:01			
Landsat 8	2 June 2013	11:14 11:14	Path 126/Row32 and Path 126/Row 33	15/30/100	
	31 May 2018	11:11	Path 126/Row 33		

**Table 3 ijerph-20-03650-t003:** Mean–standard deviation method for dividing thermal environment grades.

Thermal Environment Grade	Classification Standard
Extremely low temperature zone	T ≤ μ – 2.5 std
Low temperature zone	μ – 2.5 std < T ≤ μ – 1.5 std
Sub-low temperature zone	μ – 1.5 std < T ≤ μ – 0.5 std
Medium temperature zone	μ – 0.5 std < T ≤ μ + 0.5 std
Sub-high temperature zone	μ + 0.5 std < T ≤ μ + 1.5 std
High temperature zone	μ + 1.5 std < T ≤ μ + 2.5 std
Extremely high temperature zone	T > μ + 2.5 std

μ: the average value after normalization; std: the standard deviation after normalization.

**Table 4 ijerph-20-03650-t004:** Correlation analysis between land surface temperature and land use types at different scales.

Functional Area	Land Use Types	R	Significance	Regression Equation
Whole study area	Forest	−0.641	0.000	y = −0.057x + 40.291
	Grassland	0.231	0.021	y = 0.021x + 38.914
Mining area	Forest	−0.685	0.000	y = −0.059x + 40.061
	Opencast area	0.325	0.028	y = 0.039x + 38.890
	Dumpsite	0.380	0.009	y = 0.022x + 38.749
	Water area	−0.400	0.006	y = −1.669x + 39.254
Agricultural area	Forest	−0.681	0.000	y = −0.063x + 40.867
	Grassland	0.452	0.002	y = 0.034x + 39.114
Urban area	Forest	−0.679	0.044	y = −0.030x + 36.662

**Table 5 ijerph-20-03650-t005:** The meaning of the index that characterizes each feature.

Symbols	Feature	Index
S	Size	Reclamation area
FRAC	Shape	Fractal dimension index
*P_V_*	Reclamation mode	Vegetation coverage
Y	Reclamation term	The year since the beginning of reclamation
D	Spatial location	The distance between the gravity center of the temperature of each typical area and the gravity center of thermal effect zone of the mining area
PD	Composition of land use types around reclaimed site	The percentage of adjacent damaged land (including dumpsite and industrial site)

**Table 6 ijerph-20-03650-t006:** Correlation analysis between the cooling effect of reclaimed site and six indexes.

Parameters	Feature	T_m_	L	∆T	I
*P_V_*	Reclamation mode	−0.929 **	−0.539	0.690 *	0.515
Y	Reclamation years	−0.270	0.199	0.754 *	−0.149
S	Size	−0.396	0.436	0.188	−0.339
FRAC	Shape	0.233	0.377	−0.638 *	−0.485
D	Spatial location	−0.457	0.167	0.737 *	−0.146
PD	Composition of land use types around reclaimed site	0.363	−0.289	−0.105	0.234

* *p* < 0.05; ** *p* < 0.01.

**Table 7 ijerph-20-03650-t007:** Regression equation of cooling effect of reclaimed site and six indexes.

Y	X	Feature	R^2^	Significance	Regression Equation
T_m_	*P_V_*	Reclamation mode	0.845	0.000	y =−16.339x +46.421
∆T	*P_V_*	Reclamation mode	0.459	0.019	$y\;={\;e}^{1.702x+1.275}$
	FRAC	Shape	0.333	0.047	y =−107.092x+115.861
	D	Spatial location	0.761	0.003	$y=0.691x^{2}-8.776x+30.950$

Regression analysis was conducted for the significant correlations in Table 6.

## Data Availability

Not applicable.

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
