# Peer review of "Characteristics of the Thermal Environment and its Guidance to Ecological Restoration in a Resource-Based Area in the Loess Area"

_ijerph, 2023, doi:10.3390/ijerph20043650_

Round 1
Reviewer 1 Report
Although this topic is interesting and worthy to investigate, but in the current format, the results of this article does not have a sufficient impact to the knowledge base. The study is not original. There are too many similar studies which related remote sensing and land use.
There are some concerns and issues that are related to the clarity of the paper, as:
Introduction_Novelty unclear: What is the original contribution of the study? The introduction section is not very enlightening on the subject and the reader must read the entire manuscript to see it. Novelty should be made as clear as possible, preferably in relation with other similar studies before the manuscript is accepted for publication.
Method_ “a 2km×2km grid based on the land use classification results in 2018”, the spatial resolution is relatively low.
Conventional methods are not innovative, such as linear regression, and simply using one fitting method is not enough to support the accuracy of the results.
.
Results_Limited “The result is at a significance level of less than 0.01 and shows a good correlation between the two types of data (R2=0.424, P<0.01). Therefore, the retrieval results can reflect the distribution of the LST in the study area and can be used for further analysis.” In my opinion, R2=0.424 cannot show a good correlation, these descriptions are not rigorous enough. To test the accuracy of the result through another data, more indicators are needed
Reviewer 2 Report
The submitted manuscript deals with very important topic of heat islands, how they occur and what impacts them in relation to the land use types, in particular in case of mining areas. In planning, we talk a lot about urban heat islands, but not much about non-urban heat islands and what constitutes them and what factors affect it, and this paper contributes to understanding this phenomena.
The paper is well written with solid methodology and strong results and discussion section. The discussion section, however, deals mainly about technical aspects and I would suggest to focus on policy making in China and abroad – what are the key policy recommendations based on your research? How it can be implemented into policies in other areas in the world? This problem with heat islands and mining sites occurs in each country in the world and with raising climate change impacts, it causes a serious research area.
My second recommendation is to outline the future research needs, what should be done and how this methodology can be further utilized.
Round 2
Reviewer 1 Report
Although this topic is interesting and worthy to investigate, but in the current format, the results of this article does not have a sufficient impact to the knowledge base. There are too many similar studies which related remote sensing and land use.
